# TiO_2_-Supported Pd as an Efficient and Stable Catalyst for the Mild Hydrotreatment of Tar-Type Compounds

**DOI:** 10.3390/nano11092380

**Published:** 2021-09-13

**Authors:** Zaher Raad, Joumana Toufaily, Tayssir Hamieh, Marcelo E. Domine

**Affiliations:** 1Instituto de Tecnología Química (UPV-CSIC), Universitat Politècnica de València, Consejo Superior de Investigaciones Cientificas, Avda. de los Naranjos s/n, 46022 Valencia, Spain; zaraa@upvnet.upv.es; 2Laboratoire de Matériaux, Catalyse, Environnement et Méthodes Analytiques (MCEMA), Faculty of Sciences, Lebanese University, Hadath, Beyrouth 1003, Lebanon; joumana.toufaily@ul.edu.lb (J.T.); tayssir.hamieh@ul.edu.lb (T.H.); 3Faculty of Science and Engineering, Maastricht University, P.O. Box 616, 6200 MD Maastricht, The Netherlands

**Keywords:** tars, mild hydrotreatment, metal supported, Pd catalyst, Pd on titania

## Abstract

The mild hydrotreatment of a model mixture of tar-type compounds (i.e., naphthalene, 1-methylnaphthalene, acenaphthylene, and phenanthrene) to produce partially hydrogenated products in the range of C9–C15 was studied over Pd supported on TiO_2_ possessing different crystalline phases. Pd-based catalysts were prepared and characterized by ICP analysis, XRD, N_2_ adsorption isotherms, HR-TEM, and NH_3_-TPD, among others. The hydrotreatment activity and selectivity towards the desired hydrogenated products (i.e., tetralin and others) increased with both the acidity and surface area of the catalyst, along with the presence of small and well distributed Pd nanoparticles. For the selected 1.3 wt% Pd/TiO_2_ nano catalyst, the operational conditions for maximizing tar conversion were found to be 275 °C, 30 bar of H_2_, and 0.2 g of catalyst for 7 h. The catalyst revealed remarkable hydrotreatment activity and stability after several reuses with practically no changes in TiO_2_ structure, quite low carbon deposition, and any Pd leaching detected, thus maintaining both a small Pd particle size and adequate distribution, even after regeneration of the catalyst. Additionally, the Pd/TiO_2_ nano catalyst was demonstrated to be more active than other commonly used metal/alumina and more selective than metal/USY zeolites in the mild hydrotreatment of tar-type compounds, thus providing an efficient catalytic route for obtaining partially hydrogenated C9–C15 hydrocarbons useful as jet-fuel components or additives (improvers), as well as chemicals and solvents for industrial applications.

## 1. Introduction

In recent years, due to the continuous depletion of conventional petroleum and other fossil sources, the increasing demand for energy and high-quality transportation fuels [1], the shrinking supply of conventional crude oils [2], and the increase in petroleum prices [3] have forced petroleum refinery industries to explore heavy oils and residues to upgrade them. Thus, less popular energy sources, such as heavy oils, light cycle oil (LCO), oils or tars sands, vacuum residue (VR), bitumen, and tars, among others, have been explored and processed as alternatives [4,5]. In this context, the tars, commonly taken out from the fuel production schemes in refineries due to their refractory character, are considered one such possible fuel source.

These so-called tar feedstocks are liquid dense mixtures containing different compounds, mainly high-value mono-aromatics, such as benzenes and toluenes, among others, along with low-value poly-aromatics hydrocarbons (PAHs), such as naphthalenes, alkyl-naphthalenes, acenaphthylenes, phenanthrenes, anthracenes, and pyrenes, among others. These tar fractions are formed during petroleum distillation and/or refining, which are commonly used for the production of asphalt-type gums. In addition, they could be produced during the pyrolysis or gasification of coal or biomass [6]. In the latter case, different types of biomasses, such as industrial, agricultural and forestry residues, urban wastes, among others, could be thermochemically treated (via pyrolysis and/or gasification) to produce gases and liquids (bio-oils), which are further upgraded to obtain energy vectors, fuels, and chemicals [7,8,9,10]. In all cases, these unwanted tars condense and accumulate in reactors and pipelines during the processes, and they must be eliminated (e.g., by gas injection at high temperature or by removal with solvents) to prevent blockages, with their treatment being a challenge.

Because of their undesired above-mentioned characteristics, different refinery processes could be applied for the transformation of these heavier petroleum (or biomass-derived) fractions with high contents in mono-aromatics and mainly poly-aromatic hydrocarbons, namely light tars. For instance, feeds for the fluidized-bed cracking (FCC) unit in a refinery, such as light cycle oils (LCO), having a similar composition to light tars, are treated with solid acid catalysts (zeolites) in the FCC unit to attain lighter liquid fractions (gasoline, naphtha) together with dry gases (light alkanes, olefins, etc.) and coke [11,12]. Nevertheless, catalytic hydrotreating and hydrocracking are the most extensively explored processes for upgrading fractions containing tars [4,13,14,15,16,17,18,19,20,21,22,23]. High hydrogen pressure and elevated temperature are employed in both cases by using different types of metal supported solid catalysts. In the catalytic hydrocraking case, the presence of a metal-zeolite acid-type catalyst allows transforming tar molecules into cyclic as well as linear hydrocarbons of variable molecular weight [17,24]. In this sense, low-quality fossil tars (such as LCO and PFO) are hydroprocessed by means of catalytic hydrocracking (or selective ring opening in this case) to obtain high-quality transportation fuels (high-octane gasoline and high-cetane diesel) [13,17]. In general, the hydrotreating process is needed before hydrocracking, to obtain ultralow sulfur and nitrogen products. Moreover, these feedstocks could be transformed by catalytic hydrocracking into valuable petrochemicals, such as benzene, toluene, and xylene (BTXs enriched fractions) [13,16,19,20,21,22,23].

Several studies have been published in the literature investigating the hydrocracking of tar-type molecules into BTXs (benzene, toluene, and xylenes) by using different types of catalysts. For instance, Kim et al. studied the hydrocracking of naphthalene (at 400 °C and 30 bar of H_2_) [25] and 1-methylnaphthalene (at 380 °C and 60 bar of H_2_) [26] to attain BTXs by using Ni_2_P/Beta zeolite as catalyst, with BTXs yields of 94.4% and 42.3% for naphthalene and 1-methylnaphthalene, respectively. In addition, Upare et al. [4] worked on the hydrocracking of tetralin using CoMoS/Beta zeolite as catalyst at 370 °C and 80 bar of H_2_, thereby attaining a BTXs yield of 62.6%. The hydrocracking of 1-methylnaphthalene over a W/Beta catalyst was also studied at 420 °C and 60 bar of H_2_, yielding 53% of BTXs [27]. It is worth noting that almost the totality of the studies of tar hydrotreatment (or hydrocracking) encountered in the literature employs a single aromatic compound, thus limiting the scope of application and neglecting the competitive reactions occurring when different poly-aromatic compounds are present in the feeds. In addition, all these hydrotreatment processes mentioned above require high temperature and H_2_ pressure, with increasing energy consumption, while catalyst deactivation becomes problematic. In this sense, the processing of light tars feedstocks via catalytic hydrotreatment at lower H_2_ pressure and moderate temperature is highly recommended. In this way, heavier feedstocks having high boiling points are upgraded to more valuable medium to low boiling point products with a higher H/C ratio and higher heating value.

Bearing in mind the aforementioned reasons, the use of metal supported catalysts for tar hydrotreatments, where the support presents a lower acidity than zeolites, could be adequate to decrease the level of cracking reactions, thus increasing the amount of medium molecular weight products and avoiding the excessive production of gases and light gasoline-type compounds during the process. With this purpose, highly and homogeneously dispersed Pd nanoparticles supported on TiO_2_ material will be prepared and characterized in this study, then applied as efficient catalysts for the upgrading of a mixture of poly-aromatic hydrocarbons representative of real feedstocks of tars (derived from petroleum distillation or biomass gasification) under moderate H_2_ pressure and temperature (at 275 °C and 30 bar of H_2_). Special attention will be paid to the catalytic stability and post-reaction physico-chemical characterization, even after consecutive catalyst recycling. This mixture processing of representative tars will allow to obtain intermediate partially hydrogenated products in the range of C9–C15 cyclic hydrocarbons, by partially keeping the structure of the starting reactants. Afterwards, these intermediates could be applied as jet fuel components (or boosters) as well as valuable chemical products (i.e., solvents, reaction intermediates, etc.).

## 2. Experimental Procedure

### 2.1. Materials

SiO_2_ nanopowder (99.5%), TiO_2_ anatase (powder, 99.8%), TiO_2_ rutile (powder, 99.9%), and TiO_2_ aeroxide P25 nanopowder (99.5%) were purchased from Sigma-Aldrich (Madrid, Spain), zeolite H-Y CBV720 (molar ratio Si/Al = 15) from Zeolyst International (Kansas City, KS, USA), γ-Al_2_O_3_ (97%) from abcr (Karlsruhe, Germany), and finally TiO_2_ nanoactive (99%) and MgO nanoactive (99%) from NanoScale Corporation (Manhattan, Kansas, USA). These materials were used as supports for the preparation of Pd catalysts, after a soft treatment at 100 °C to remove water. In addition, commercial 1 wt% Pd/Al_2_O_3_ and 1 wt% Pt/Al_2_O_3_ catalysts were purchased from Sigma-Aldrich (Madrid, Spain).

Pd(NH_3_)_4_Cl_2_∙H_2_O (99.99%), Pd(NO_3_)_2_∙2H_2_O (40% Pd basis), Pd(NO_3_)_2_∙4NH_3_, and Pt(NH_3_)_4_Cl_2_∙xH_2_O (99.99%) were purchased from Sigma-Aldrich (Madrid, Spain) and used as received as metal precursors for the preparation of metal supported materials, by diluting them in the corresponding aqueous solutions (Water: Milli-Q quality, Millipore).

For the preparation of the reaction model mixture, the following reactants were used: naphthalene (99%) and phenanthrene (98%) provided by Sigma-Aldrich (Madrid, Spain), n-hexadecane (99%) anhydrous, AcroSeal from Acros Organics (Geel, Belgium), acenaphthylene (>90%) provided by TCI Europe (Zwijndrecht, Belgium), and 1-methylnaphthalene (96%) from Alfa Aesar (Kandel, Germany). For the dilution of liquid samples to perform chromatographic analysis, 1 wt% of n-decane (Sigma-Aldrich, 99%) (Madrid, Spain) dissolved in Tetrahydrofuran, THF (Sigma-Aldrich, 99%) (Madrid, Spain) was used as standard solution.

### 2.2. Catalyst Preparation

Pd supported on monometallic oxides catalysts were prepared by the incipient wetness impregnation method [28] using Pd(NH_3_)_4_Cl_2_∙H_2_O. Briefly, an adequate amount of Pd precursor was dissolved in milli-Q water, according to the final desired Pd content. Then, this solution was added dropwise to the support under continuous mixing until the formation of a homogeneous gel. Afterwards, the material was dried at 100 °C for 3 h to remove water, then reduced at 400 °C (3 °C/min) for 2 h under a H_2_ flow of 100 mL/min prior to its use in the catalytic test. Samples were prepared with Pd loadings between 0.8 and 2.2 wt%. Different Pd precursors, such as Pd(NO_3_)_2_∙2H_2_O and Pd(NO_3_)_2_∙4NH_3_, were also used for comparative purposes.

### 2.3. Catalysts Characterization

X-ray diffraction (XRD) was used to determine the crystalline phase for the different TiO_2_ materials, along with the presence of Pd^0^ species. XRD patterns were performed using a PANalytical Cubix Diffractometer (Cu kα radiation and graphite monochromator) (PANalytical, Almelo, The Netherlands).

Measurements of physical adsorption of nitrogen at 77 K using a Micromeritics ASAP 2420 instrument (Micromeritics, Norcross, GA, USA) were performed to collect the N_2_ adsorption isotherms. Samples (200–300 mg with a granulometry between 0.4–0.8 mm) were degassed in a vacuum overnight at 350 °C. In addition, surface areas of the materials were calculated using BET method.

The total amount of acid sites for the various Pd/TiO_2_ catalysts was determined by temperature-programmed desorption of ammonia (NH_3_-TPD) carried out using a TPD/2900 apparatus from Micromeritics (Norcross, GA, USA). Typically, a pre-treatment at 450 °C of the sample (100 mg with a granulometry between 0.4–0.8 mm) was performed in an Ar stream for 1 h for the adsorbed contaminants and volatile species removal. Afterwards, NH_3_ was chemisorbed by pulses at 100 °C until attaining equilibrium. Then, the sample was fluxed with helium stream for 15 min to remove the excess of NH_3_, prior to increasing the temperature up to 500 °C in a 100 mL/min of He stream at a heating rate of 10 °C/min. NH_3_ desorption was scanned with a thermal conductivity detector (TCD) (Micromeritics, Norcross, GA, USA) and a mass spectrometer (Micromeritics, Norcross, GA, USA).

Pd loading was determined by using an ICP inductively coupled plasma emission spectrophotometer (Varian 715-ES) (Agilent Technologies, Santa Clara, CA, USA). The amount of organic matter deposited onto the solids after reaction was quantified by means of elemental analyses in an Euro EA 3000 Elemental Analyzer (EUROVECTOR, Milan, Italy).

Pd metal dispersion and average Pd nanoparticle size of the sample were estimated from CO adsorption using the double isotherm method using Quantachrome Autosorb-1C equipment (Quantachrome Instruments, Boynton Beach, FL, USA). Prior to adsorption, 300 mg of the catalyst (0.4–0.8 mm) were reduced under H_2_ flow by using the same reduction temperature applied for catalysts (at 400 °C during 2 h and 3 °C/min). Afterwards, samples were degassed at 1333 × 10^−3^ Pa for 2 h at 400 °C, and then the temperature was decreased to 35 °C. Next, pure CO was admitted and the first adsorption isotherm (i.e., the total CO uptake) was measured. After evacuation at 35 °C, the second isotherm (i.e., the reversible CO uptake) was taken. The amount of chemisorbed CO was determined by subtracting the two isotherms. The pressure range studied was 0.5–11 × 10^4^ Pa. Pd dispersion was calculated from the amount of irreversibly adsorbed CO, supposing a stoichiometry Pd/CO = 1.

TPR (temperature-programmed reduction) analysis was carried out using Micromeritics Autochem 2910 equipment (Micromeritics, Norcross, GA, USA). Calcined catalysts were initially flushed with 30 cm^3^/min of Ar at room temperature for 30 min and then a mixture of 10 vol% of H_2_ in Ar was passed through the catalyst at a total flow rate of 50 cm^3^/min while the temperature was increased to 800 °C at a heating rate of 10 °C/min. The H_2_ consumption rate was monitored in a thermal conductivity detector (TCD) (Micromeritics, Norcross, GA, USA), previously calibrated using the reduction of CuO as reference.

X-ray photoelectron spectroscopy (XPS) data were obtained from a SPECS spectrometer equipped with a 150-MCD-9 detector (SPECSGROUP, Berlin, Germany) and using a non-monochromatic Al K (1486.6 eV) X-ray source. Spectra were recorded at 200 °C, using an analyzer pass energy of 30 eV, an X-ray power of 50 W, and under an operating pressure of 10^−9^ mbar. During data processing of the XPS spectra, binding energy (BE) values were referenced to the C1s signal. Spectra treatment has been performed using the CASA software (version 2.3.16Dev52, Casa software Ltd., Teignmouth, United Kingdom).

For textural characterization of the solids, transmission electron microscopy (TEM) (JEOL, Akishima, Tokyo, Japan) measurements were carried out using JEOL JEM-1010 instrument operating at 200 KV, equipped with a digital camera (Mega View III) (DEBEN UK Ltd., Suffolk, United Kingdom). In addition, high-resolution transmission electron microscopy (HR-TEM) measurements were performed using JEOL JEM-2100F equipment (JEOL, Akishima, Tokyo, Japan), working at a voltage of 200 kV. Pd dispersion was checked by X-ray-energy-dispersive spectroscopy to obtain the elemental mapping using a JEOL 6300 scanning electron microscope (SEM) equipped with an Oxford LINK ISIS detector (Oxford Instruments, United Kingdom). Mapping images were obtained with a focused beam of electrons (20 KV) and a counting time of 50–100 s.

### 2.4. Catalytic Experiments

Catalytic hydrotreatment reactions were carried out in a homemade 12 mL autoclave-type reactor with interior vessel of PEEK (Polyether ether ketone) reinforced with graphite, equipped with a magnetic stirrer bar, pressure control, and a valve for either liquid or gas sample extraction at different time intervals. The reactor was placed over a steel jacket individual support connected to a heater programmer to control the temperature. Typically, 0.5 g of a model mixture representative of tars composed by naphthalene (0.125 g), acenaphthylene (0.125 g), phenanthrene (0.125 g), and 1-methylnaphthalene (0.125 g) in 4.0 g of n-hexadecane anhydrous solvent, together with 0.20 g of catalyst added to the reactor, which was then hermetically closed. The reactor was first purged with N_2_, then pressurized at 10–36 bar of H_2_ and heated at 250–300 °C under continuous stirring (1000 rpm). This reaction continued for 7 h. Small liquid aliquots (≈100 µL) were collected at different time intervals, diluted with 0.5 g of 1 wt% n-decane in Tetrahydrofuran (THF) standard solution, then centrifugated to remove solid the catalyst prior to chromatographic analysis. Liquid samples were analyzed by an Agilent Technologies 7890A GC system with an FID detector (Agilent Technologies, Santa Clara, CA, USA) equipped with a HP-5 MS capillary column (30 m × 250 µm × 0.25 µm). The products were identified by Agilent 6890 N GC system coupled with an Agilent 5973 N mass detector (Agilent Technologies, Santa Clara, California, USA) and equipped with the same capillary column.

Recycling experiments of the Pd/TiO_2_ nanoactive catalyst were performed by using the solid catalyst in the reaction under the standard operational conditions (first use), recovering it from the reaction mixture, washing the latter with 2-propanol with centrifugation. Then, the solid was dried at 100 °C for 1 h and finally reused in a new reaction. This process was repeated two more times. After the third reuse, the catalyst was recovered and regenerated by thermal treatment (see Section 2.2).

Naphthalene, 1-Methylnaphthalene, Phenanthrene (Phe) and Acenaphthylene (Acy) conversion and products selectivity (according to GC datas) were calculated as follows:Conversion (%) = [(moles of reactants (t_0_) − moles of reactants (t))/moles of reactants (t_0_)] × 100
Selectivity (%) = [moles of product (t)/(moles of reactants (t_0_) − moles of reactants(t))] × 100

The Turn Over Number (TON) was defined as:TON = moles of products (t)/moles of metal in catalyst

The mass balance is defined as:Mass balance (%) = 100 × (mass products + reactants(t))/mass of reactants(t_0_)

In order to facilitate the analysis of reaction mixtures and discussion of the results, the liquid products obtained in reaction were grouped as depicted in Figure 1. The different groups are as follows: MonoAr “monoaromatics and/or alkyl-monoaromatics”, tetralin “tetralin and methyl-tetralin”, decalin “cis/trans decalin and methyl-decalin”, Ace “Acenaphthene”, HAce-1 “1 hydrogenated ring of acenaphthene”, HPhe-1 “1 hydrogenated ring of phenanthrene (1,2,3,4-tetrahydrophenanthrene, TetHPhe, and 9,10-dihydrophenanthrene, DiHPhe)”, HPhe-2 “2 hydrogenated rings of phenanthrene (1,2,3,4,4a,9,10,10a-octahydrophenanthrene, asymOHPhe, and 1,2,3,4,5,6,7,8-octahydrophenanthrene, symOHPhe)”, and HAce-2/HPhe-3 “totally hydrogenated products of acenaphthylene/phenanthrene (perhydrophenanthrene), respectively”.

## 3. Results and Discussion

### 3.1. First Catalytic Screening

In a first attempt, and in order to assess the adequate metal selection for the mild hydrotreatment of tars, different metals supported on carbon commercial catalysts (Ru/C, Pt/C, and Pd/C) were essayed in the reaction. These preliminary results (see Appendix A) showed that the Pd/C catalyst achieved better levels of conversion of tar-type reactants and higher hydrogenation activity than the analogous Ru and Pt supported on carbon materials. Consequently, Pd was selected as the appropriate active metal phase for the mild hydrotreatment of tar-type compounds. Afterwards, different Pd nanoparticle-based materials were prepared by depositing ≈ 2 wt% Pd (through incipient wetness impregnation method) onto different commercial metal oxides, such as TiO_2_, γ-Al_2_O_3_, MgO, and SiO_2_. (X-rays patterns of Pd-based catalysts in Appendix A). The Pd-based catalysts listed in Table 1 were screened in the mild hydrotreatment of tar-type molecules at 250 °C and 30 bar of H_2_ pressure for 7 h. As can be seen in Table 1 (and Appendix A), the highest levels of both conversion (89%) and TON (67) were achieved with Pd/TiO_2_ nano catalyst. Despite the lower surface area of TiO_2_ nano compared to the rest of the supports used, the latter catalyst showed better activity than Pd/γ-Al_2_O_3_ catalyst, and much better than Pd/MgO and Pd/SiO_2_ (with 63 and 54% conversion, respectively). The selectivity towards the different groups of products detected in the reaction mixture is presented in Figure 2. In general, the more active catalyst is expected to show higher selectivity to tetralin, decalin, HPhe-2, and HAce-2/HPhe-3 types of product, and lower selectivity to monoAr, Ace, and HPhe-1, respectively. For instance, when the selectivity of all the Pd-based catalysts of Table 1 was compared in the range of 54–57% conversion, Pd supported on TiO_2_ nano and Al_2_O_3_ catalysts, also followed by MgO sample, offered quite similar selectivities to the different groups of products, whereas Pd/SiO_2_ was found to be slightly less selective to tetralin and more selective to acenaphthene, Ace. In this sense, the superior activity of Pd/TiO_2_ nano (followed by Pd/Al_2_O_3_) compared to other metal oxides supported-Pd catalysts could be mainly due to the adequate acidity of the materials allowing reactants and support to interact properly, accompanied by an adequate hydrogenating capacity facilitated by the interaction between the small Pd nanoparticles and the support. In addition, the slightly better behavior of Pd/TiO_2_ nano vs. Pd/Al_2_O_3_ could be related to the more adequate acidity (ratio of Brönsted/Lewis acid sites) in the former catalyst, thus reducing the strong adsorption of intermediates and products onto the catalyst surface and avoiding the consequent C deposition (responsible of the catalyst deactivation). Taking into consideration the above, Pd/TiO_2_ nano was found to be the most efficient catalyst in the mild hydrotreatment of tar-type compounds, and thus it was selected for further studies.

In addition, a rationalized reaction scheme for this model tars-compounds mild hydrotreatment is proposed in Figure 3. One aromatic ring of the naphthalene molecule (and methylnaphthalene) is hydrogenated to produce tetralin (and methyl-tetralin), which is either further converted into totally hydrogenated cis/trans decalin (and methyl-decalin) or suffers C-C bond cleavage through cracking reaction (H^+^), and then alkylbenzene (MonoAr) is obtained as minor product. Additionally, the non-aromatic double bond of acenaphthylene (Acy) is firstly hydrogenated to produce acenaphthene (Ace), and afterwards the latter is transformed into partially and totally hydrogenated products (1-hydroacenaphthene HAce-1 and 2-hydroacenaphthene HAce-2, respectively). With respect to phenanthrene, one aromatic ring is hydrogenated to produce the corresponding primary products DiHPhe and TetHPhe (HPhe-1 group). The latter are further converted into HPhe-2: symOHPhe (from TetHPhe) and asymOHPhe (from both DiHPhe and TetHPhe). Finally, perhydrophenanthrene HPhe-3 is formed, which is a totally hydrogenated product of phenanthrene.

### 3.2. Effect of Pd Content in Pd/TiO_2_ Catalyst

Pd/TiO_2_ nanocatalysts with different metal contents (0.8 wt%, 1.3 wt%, and 2.2 wt%, respectively) were prepared and characterized by different techniques (ICP, XRD, TEM), then tested in the mild hydrotreatment of tar-type molecules to ascertain the optimal Pd loading needed in the solid catalyst.

For instance, X-ray diffraction patterns of the pure TiO_2_ nano support (Figure 4) presented weak anatase peaks, whereas X-rays diffraction patterns of the as prepared Pd/TiO_2_ nano samples predominantly showed the presence of the anatase phase and a small amount of brookite. The difference in the anatase crystallinity between pure TiO_2_ and Pd/TiO_2_ could be due to the activation of the Pd-based catalyst with H_2_ at 400 °C. In addition, the intensity of the diffraction peaks attributed to Pd^0^ species increased when increasing the Pd content in the solid from 0.8 wt% to 2.2 wt%. This tendency is also in agreement with the increase in the Pd nanoparticle sizes (from 4–7 up to 12 nm) observed when increasing the Pd loadings in the solids (see Appendix A). Noticeably, these Pd-based catalysts presented very similar diffraction patterns when other Pd precursors were used for their preparation (data not shown), while the catalytic activity remained practically unchangeable (see Appendix A). These results evidenced that the Pd precursor was not a key point for the synthesis of catalysts.

The Pd/TiO_2_ nano catalysts were evaluated in the mild hydrotreatment of tar compound model mixture (at 250 °C and 30 bar of H_2_ with 0.2 g of catalyst during 7 h), and the results in terms of attained tar conversion and calculated TON are summarized in Appendix A). As can be seen, the highest tar conversion (≈90% at 7 h) was encountered for 2.2 wt% Pd/TiO_2_ sample followed by 1.3 wt% Pd/TiO_2_ (Conv. ≈75%) and 0.8 wt% Pd/TiO_2_ (Conv. ≈63%) catalysts, respectively. On the contrary, calculated TON values followed the reverse order: 0.8 wt% Pd/TiO_2_ (TON = 117) > 1.3 wt% Pd/TiO_2_ (TON = 91) > 2.2 wt% Pd/TiO_2_ (TON = 67). For the two latter Pd-based catalysts, a quite similar selectivity to the different groups of products was encountered at 63–65% range of conversion, while 0.8 wt% Pd/TiO_2_ catalyst showed a slightly lower selectivity to tetralin and HPhe-2 hydrogenated products (see Appendix A). From these results, it can be concluded that 1.3 wt% Pd/TiO_2_ nano material presents a good compromise between Pd loading, catalytic activity (tars conversion and TON), and selectivity to the different groups of products. Consequently, the 1.3 wt% Pd loaded TiO_2_ material was selected for further studies.

### 3.3. Effect of TiO_2_ Crystalline Phase Used as Support

Different titanium oxide samples presenting different crystalline phases, such as TiO_2_ nano, TiO_2_ P25, TiO_2_ anatase, and TiO_2_ rutile were used for the preparation of Pd/TiO_2_ type catalysts (XRD patterns in Appendix A) aiming to check the effect of support crystalline phases on the catalytic performance of the catalysts in the mild hydrotreatment of tars. Commercial TiO_2_ nano is a high surface area nanocrystalline sample majorly composed of an anatase phase of titania, while TiO_2_ P25 is a widely used titanium oxide sample composed of a mixture of anatase and rutile phases of titania. In addition, TiO_2_ anatase and TiO_2_ rutile are pure commercial samples of titania anatase and rutile phases, respectively. As seen in Appendix A, a Pd(111) diffraction peak was detected in all Pd-based catalysts. Table 2 shows the main textural and physico-chemical properties, as well as the catalytic performance of the above-mentioned ≈1.3 wt% Pd/TiO_2_ materials (see also Appendix A). As can be seen, catalytic activity order in terms of both tar conversion and TON encountered for these samples was: Pd/TiO_2_ nano > Pd/TiO_2_ P25 > Pd/TiO_2_ anatase > Pd/TiO_2_ rutile. This tendency correlates with the surface areas measured for each one of the supports, as well as with the metal dispersion and metal particle size determined by CO chemisorption (see Table 2). Thus, Pd supported onto TiO_2_ nano material possesses the smallest Pd particle size (13 nm), the highest Pd dispersion (Pd/TiO_2_ P25 as well), along with the highest catalytic activity (Conv. ≈75%) towards tar conversion in the mild hydrotreatment process compared to the other TiO_2_-supported Pd catalysts. For instance, Pd/TiO_2_ nano was demonstrated to be a more active and selective catalyst towards hydrogenated products, e.g., tetralin group, compared with the other Ti-based catalysts (see Figure 5, selectivity to different products at 35–40% of conversion). Interestingly, the selectivity to hydroacenaphthene (HAce-1), which is a more hydrogenated secondary product derived from primary product acenaphthene (Ace), increased in the following order: Pd/TiO_2_ rutile (16%) < Pd/TiO_2_ anatase (23%) < Pd/TiO_2_ P25 (32%) < Pd/TiO_2_ nano (38%). This clearly suggests that Pd/TiO_2_ nano was more active to hydrogenate the ring of Ace to obtain HAce-1 than the other TiO_2_ type catalysts. Therefore, these data let us conclude that the Pd/TiO_2_ nano catalyst is the more active and selective towards the hydrogenated products among the different TiO_2_-supported Pd catalysts tested here.

The acidic properties of the selected TiO_2_-supported Pd catalysts were studied through TPD-NH_3_ measurements, and the amounts of adsorbed ammonia representing the total amount of acid sites determined for each sample are listed in Table 3. For instance, the total amounts of acid sites encountered for Pd/TiO_2_ nano and Pd/TiO_2_ P25 were 367 and 200 µmol/g, respectively. Both catalytic samples presented a broad distribution of the ammonia adsorption–desorption peak from 100–500 °C. Thus, whereas Pd/TiO_2_ P25 showed only one broad peak at 250 °C, Pd/TiO_2_ nano showed two peaks at 250 °C and 350 °C, respectively (see Appendix A). Noticeably, these two peaks for Pd/TiO_2_ nano were distributed in two regions, related to weaker acid sites for the peak at lower temperature (250 °C) and stronger acid sites for the peak at higher temperature (350 °C). However, the very low interaction of ammonia with Pd/TiO_2_ anatase and Pd/TiO_2_ rutile was detected (practically no peaks observed in Appendix A), leading to low values of the total number of acid sites appearing in Table 3 (43 and 23 µmol/g, respectively). Summarizing, TiO_2_ nano and TiO_2_ P25 were found to be the more acidic samples, also having the higher surface areas, and both properties combined improved their catalytic properties and the conversion of tar-type compounds. Thus, higher conversion was observed for Pd/TiO_2_ nano followed by TiO_2_ P25 catalyst. More importantly, the higher selectivities towards the more hydrogenated products (i.e., tetralin, HAce-1, etc.) were also observed with these two catalysts (see Figure 5).

Additionally, and aiming at investigating the reducibility and hydrogenating capacity of Pd species, H_2_-TPR experiments of the different TiO_2_-supported Pd catalysts were performed and the total amounts of adsorbed H_2_ in each case are reported in Table 3 (see also TPR profiles in Appendix A). For Pd/TiO_2_ nano and Pd/TiO_2_ P25 samples, a band with maximum centered at 378 °C and 392 °C, respectively, was detected and their corresponding H_2_ uptakes (193 and 115 µmol/g, respectively) were calculated. On the contrary, practically no H_2_ adsorption peaks were detected for Pd/TiO_2_ rutile and Pd/TiO_2_ anatase. Therefore, negligible hydrogen consumption occurred in both Pd/TiO_2_ anatase and Pd/TiO_2_ rutile samples. Moreover, TPR profiles (Appendix A) showed a negative peak (corresponding to the decomposition of Pd hydrides PdHx species) at 63 °C for all the samples, although the intensity of this peak changed depending on the support type. In general, this type of species is formed through H_2_ adsorption/diffusion in the Pd^0^ crystallites at lower temperature [29]. From the H_2_-TPR data here exposed, Pd/TiO_2_ nano and Pd/TiO_2_ P25 appear to be more capable to adsorb and dissociate H_2_ at the catalyst surface than the other analogous Pd/TiO_2_ samples, these results correlate pretty well with the catalytic activities observed (see Table 2). This major hydrogenating capacity of Pd/TiO_2_ nano followed by Pd/TiO_2_ P25 is probably due to the minor metal particle size and the higher metal dispersion (also related to the higher surface areas of TiO_2_ nano and TiO_2_ P25 supports), which increase the adequate interaction between Pd species and TiO_2_ support.

In light of all the above-mentioned, and mainly taking into consideration the physico-chemical (acidity, surface area, Pd nanoparticles sizes, etc.) and catalytic properties demonstrated, Pd/TiO_2_ nano was selected as the adequate catalyst for the mild hydrotreatment of tar-type compounds, and further studies were performed aiming to optimize its usage in this process.

### 3.4. Effect of Reaction Conditions for Pd/TiO_2_ Nano Catalyst

Optimization of the operational conditions for the mild hydrotreatment of tars for a 1.3 wt% Pd/TiO_2_ nano catalyst was performed by studying the effect of reaction parameters, such as temperature, H_2_ pressure, and amount of catalyst, on the conversion of tar-type compounds and selectivity towards the more hydrogenated products. On one hand, reactions at 250 °C, 275 °C, and 300 °C were carried out to evaluate the influence of temperature on the catalytic activity of Pd/TiO_2_ nano catalyst. As shown in Figure 6A, the conversion increased from 75% at 250 °C to ≈90% at 275 °C, and no significant difference was detected with further increases of the temperature till 300 °C. Meanwhile, as depicted in Figure 7, moderate to high selectivity to tetralin and other more hydrogenated product groups (i.e., HPhe-2) was found when increasing the temperature from 250 °C to 275 °C (and 300 °C). On the other hand, the influence of the H_2_ pressure on the catalytic performance of the 1.3 wt% Pd/TiO_2_ nano was investigated at 275 °C, using 0.2 g of catalyst for 7 h. The reactions were carried out at 10, 20, 30, and 36 bar of H_2_. As shown in Figure 6B, the conversion increased from 54% to ≈90% when increasing the H_2_ pressure from 10 to 30 bar, while practically similar conversions were observed at both 30 and 36 bar of H_2_. This is probably due to H_2_ solubility limitations in the n-hexadecane under the reaction conditions, along with the autoclave-type reactor used here (see Experimental section). In addition, as seen in Appendix A, similar selectivities towards the different groups of products (compared at the same level of conversion) were found when working at 275 °C and H_2_ pressure ≥20 bar. However, higher selectivity to the less hydrogenated primary product acenaphthene (Ace) was obtained when working at 10 bar of H_2_.

From all these results, the optimal operational conditions to carry out the mild hydrotreatment of tars over 1.3 wt% Pd/TiO_2_ nano were 275 °C and 30 bar of H_2_ pressure. Under these reaction conditions, catalyst loading was also optimized by performing experiments with catalyst ranging from 0.05 to 0.250 g (see Appendix A), with the maximum conversion being achieved by working with 0.2 g of 1.3 wt% Pd/TiO_2_ nano catalyst.

### 3.5. Reusability Tests

In general, one important point concerning the usage of metal-supported type catalysts for the mild hydrotreatment of tar-type molecules depends on the possibilty of recycling the solid catalyst several times. In order to evaluate the stability of the studied 1.3 wt% Pd/TiO_2_ nano catalyst under reaction conditions and its remaining activity after several reuses, a set of experiments was performed in which the Pd-based catalyst used in a first reaction was recovered and then recycled at least three more times. Each time, the spent catalyst was recuperated from the reaction mixture at the end of the experiment by centrifugation, then washed with 2-propanol and dried at 100 °C during 1 h, before its use in a new catalytic experiment (see Experimental section). Results obtained from reusability tests for Pd-supported on TiO_2_ nano catalysts in the mild hydrotreatment of tars are summarized in Table 4 and Figure 8.

As can be seen, the catalytic activity evaluated in terms of tars-type compounds conversion at three different reaction times (1, 3, and 7 h) did not suffer any decay, even after four consecutive uses of the catalyst (Figure 8). However, after the first use, a small amount of carbon was deposited on the catalyst surface, and its percentage increased after three more uses without affecting the catalytic activity. It is worth noting that after catalyst regeneration at 400 °C under H_2_ flow, the carbon deposition strongly decreased (from ≈1.5 to ≈0.6, see Table 4). In addition, any Pd leaching was detected after several reuses (Table 4). In fact, a small particle size was encountered with an average diameter around 2 nm for Pd/TiO_2_ fresh, reused, and regenerated catalysts (see Table 4 and Appendix A). In this sense, although the fresh catalyst showed most Pd particle sizes between 1.0 and 1.5 nm, there was a certain number of particles with sizes higher than 2 nm, leading to an average particle size of around 2.7 nm for the fresh material, which evidences that no increase in the particle size occurred after several reuses. This can also be observed from the TEM images of Figure 9 (and Appendix A), where practically no changes in Pd particles sizes and their distribution were observed between fresh, used, and regenerated catalysts.

The X-ray diffractions patterns after first and fourth uses of the Pd/TiO_2_ nano catalyst (Figure 10) showed practically identical signals for all the measured catalytic samples, with only a small shift in the peak placed at 2θ = 40° assigned to Pd(111), probably due to the transformation of Pd^0^ into Pd^2+^ species. This indicates that the structure of the catalyst remained practically unaltered and just a few changes in the Pd species state take place during the hydrotreatment process.

In order to gain new insights regarding the role of Pd species in the catalyst and aiming to understand what was occurring on the solid surface during the reaction, X-ray photoelectron spectroscopy (XPS) analysis was employed to evaluate the electronic properties and the nature of Pd species present before and after reuses in the Pd/TiO_2_ nano catalyst. Results of XPS measurements carried out over fresh, reused, and regenerated (with H_2_) catalyst are presented in Figure 11 and Table 5. It was found that Pd 3d spectra of the fresh catalyst can be divided into four components located at 334.64 eV (Pd^0^ 3d_5/2_), 336.48 eV (Pd^2+^ 3d_5/2_), 339.87 eV (Pd^0^ 3d_3/2_), and 341.84 eV (Pd^2+^ 3d_3/2_). For the reused sample, two components were found located at 335.71 eV (Pd^2+^ 3d_5/2_) and 340.96 eV (Pd^2+^ 3d_3/2_), respectively. Finally, another four components located at 335.07 eV (Pd^0^ 3d_5/2_), 337.05 eV (Pd^2+^ 3d_5/2_), 340.35 eV (Pd^0^ 3d_3/2_), and 342.05 eV (Pd^2+^ 3d_3/2_) were encountered in the Pd 3d spectra of the regenerated catalyst. From these data, the corresponding relative abundance of Pd species present in the catalytic samples was calculated and the results are exposed in Table 5. As can be seen, the fresh catalyst presented a Pd^0^/Pd^2+^ species ratio of around 80/20, while after the catalyst usage in the reaction only the presence of Pd^2+^ species (100%) was detected in the solid. Nevertheless, after regeneration with the H_2_ of the Pd/TiO_2_ nano catalyst, the Pd^0^ species were recovered and the Pd^0^/Pd^2+^ species ratio was reestablished. In fact, this Pd^0^/Pd^2+^ ratio was higher in the regenerated catalyst than in the fresh material.

All these results allow us to conclude that the Pd^0^ species initially present in the Pd/TiO_2_ nano catalyst are responsible for the catalytic activity demonstrated in the hydrotreatment process. Nevertheless, these Pd^0^ species are transformed to Pd^2+^ species during the process, these species being the only ones present in the used catalyst. Therefore, in order to answer the question pertaining to how the used catalyst is capable of maintaining its catalytic activity after several reuses, an experiment involving the mild hydrotreatment of tars was performed using a specifically prepared PdO/TiO_2_ nano catalyst working at 275 °C and 30 bar of H_2_ (optimal conditions). The attained results were very similar to those obtained with the Pd/TiO_2_ nano catalyst reduced (activated under H_2_ flow) prior to the reaction. This meaning that the catalyst is able to be activated or “in situ” reduced under reaction conditions. This fact could explain the maintained catalytic activity after reuses, where the catalyst was “in situ” reactivated with H_2_ and Pd^2+^ species formed during the hydrotreatment process were transformed into Pd^0^ (redox cycle) under these reaction conditions, which confirmed the stability and the reusability of the catalyst.

These results let us propose a probable action mechanism of the Pd/TiO_2_ nano catalyst in the mild hydrotreatment of tars. Thus, it is plausible to conclude that Pd^0^ species are responsible for the H_2_ activation and dissociation (H---H), while the support (due to its adequate acidity) favors the adsorption of reactants close to the Pd active center, thus leading to the hydrogenation of C=C bounds of unsaturated molecules. In the meantime, Pd metallic species are transformed into PdO, which in the presence of H_2_ and assisted by the metal-TiO_2_ support interaction are again reduced to Pd^0^, with the redox cycle being performed successfully. Nevertheless, when C deposition occurs during the process, this metal–support interaction is reduced or inhibited, thus avoiding the PdO to Pd^0^ conversion and leading to catalyst deactivation.

Summarizing, all these data revealed the remarkable stability of the Pd/TiO_2_ nano catalyst in the mild hydrotreatment of tars, which can be efficiently reused several times with practically no changes in TiO_2_ structure, quite low carbon deposition, and any Pd leaching detected to maintain both small Pd particle sizes and their adequate distribution, even after regeneration of the catalyst.

### 3.6. Comparison with Commercial and Previously Reported Catalysts

Up to now, the Pd/TiO_2_ nano material here studied has been demonstrated to be an efficient and stable catalyst in the mild hydrotreatment of tar-type compounds for obtaining hydrogenated and partially hydrogenated hydrocarbons in the C9–C15 range. Of course, a wide range of metal-supported catalysts have been used for the hydrotreatment of tar molecules. Therefore, a comparison of our Pd/TiO_2_ nano material with some selected and previously reported hydrotreatment catalysts under the reaction conditions employed in this work was performed and the attained results are summarized in Table 6. As can be seen, our Pd/TiO_2_ nano catalyst was more active (higher conversion and comparable TONs) than Pd/Al_2_O_3_ and Pt/Al_2_O_3_ catalysts, also showing better selectivity to the more hydrogenated products (tetralin and others). With respect to CoMo- and NiMo-based commercial catalysts, Pd/TiO_2_ nano showed higher conversion than that of CoMoS/SiO_2_-Al_2_O_3_ and comparable to that observed with NiMoS/Al_2_O_3_. However, Pd/TiO_2_ nano was found to be more active having a higher TON (103) than that calculated for both NiMo- and CoMo-based samples (TON = 7), along with less carbon deposition on the catalyst surface after reaction. In addition, although the conversion achieved with Pd and Pt supported on zeolite USY was higher than in the case of Pd/TiO_2_ nano, significant differences in the mass balance values and carbon deposition were detected. In the same way, zeolite-supported materials showed different selectivity compared to Pd/TiO_2_ nano and NiMoS/Al_2_O_3_, which were encountered to be selective to the same type of hydrogenated or partially hydrogenated products. For instance, as mentioned in Table 6, Pd/USY and Pt/USY showed higher selectivity to monoaromatics (including BTXs in this case) than Pd/TiO_2_ nano, and lower selectivity to the hydrogenated or partially hydrogenated products, such as tetralin, HPhe-1, and HPhe-2. In addition, from the mass balance values and the higher selectivity to some cracked products, it could be concluded that higher amounts of gases were produced over Pd/USY and Pt/USY, along with some other by-products (i.e., alkylated and dialkylated aromatics), which remain beyond the scope of this specific study. Thus, despite the higher activity of metal-zeolites (mainly due to their hydrocracking properties), the high production of gases and coke (deposited on the catalyst surface), along with the low production of hydrogenated products become important disadvantages when the production of partially hydrogenated C9–C15 range hydrocarbons are the targeted products.

## 4. Conclusions

The mild hydrotreatment of a model mixture of tar-type compounds (i.e., naphthalene, 1-methylnaphthalene, acenaphthylene, and phenanthrene), simulating those produced from petroleum distillation or from biomass gasification, into hydrogenated and partially hydrogenated products in the range of C9–C15 was studied over Pd supported on TiO_2_ catalysts. The hydrotreatment activity and selectivity towards the desired hydrogenated products (i.e., tetralin and others) were strongly dependent on the number of acid sites and the surface area of the catalysts, together with Pd particle size and their proper distribution. Thus, the increase in both the acidity and surface area of the catalyst, along with the presence of small and well distributed Pd nanoparticles, lead to an improvement of the activity for the mild hydrotreatment of tars. Among different TiO_2_ crystalline phases used as support, TiO_2_ nano possessing mainly titania anatase phase was found to be the more adequate to accommodate small Pd nanoparticles. For the selected 1.3 wt% Pd/TiO_2_ nano catalyst, the operational conditions found to maximize both conversion and selectivity to the desired products were: 275 °C, 30 bar of H_2_, and 0.2 g of catalyst for 7 h. More interestingly, after consecutive reuses, the Pd/TiO_2_ nano catalyst remained active and stable, with very low carbon deposition, while any Pd leaching was detected, and there were practically no changes in the Pd nanoparticle size, even after regeneration (with H_2_) of the used catalyst. Thus, although Pd^0^ active species were transformed into Pd^2+^ species during the hydrotreatment process, the Pd metallic species were recovered when the used catalyst was added to the reaction medium for a new test via an “in situ” reduction, similarly to what could be achieved by “ex situ” regeneration under H_2_ flow. Finally, the activity of Pd/TiO_2_ nano catalyst in the mild hydrotreatment of tars was discussed in comparison to other previously reported hydrotreating catalysts, such as CoMoS/SiO_2_-Al_2_O_3_ and NiMoS/Al_2_O_3_, as well as metal supported on alumina and zeolite H-USY. For instance, the Pd/TiO_2_ nano catalyst was demonstrated to be more active and selective than metal supported on alumina and CoMo-based catalysts, with conversion values comparable to those of NiMoS/Al_2_O_3_, but having much a higher TON and less carbon deposition. Additionally, other types of products (higher amounts of gases, light hydrocarbons and monoaromatics) were generated with Pd/USY as catalyst, along with high carbon deposition on the catalyst surface. In summary, the Pd/TiO_2_ nano catalyst was found to be an efficient and stable catalyst for the mild hydrotreatment of tar-type compounds to obtain hydrogenated and partially hydrogenated C9–C15 hydrocarbon products that could be applied as fuels components or additives (i.e., jet-fuel improvers), or as chemicals and solvents for industrial applications.

## Figures and Tables

**Figure 1 nanomaterials-11-02380-f001:**
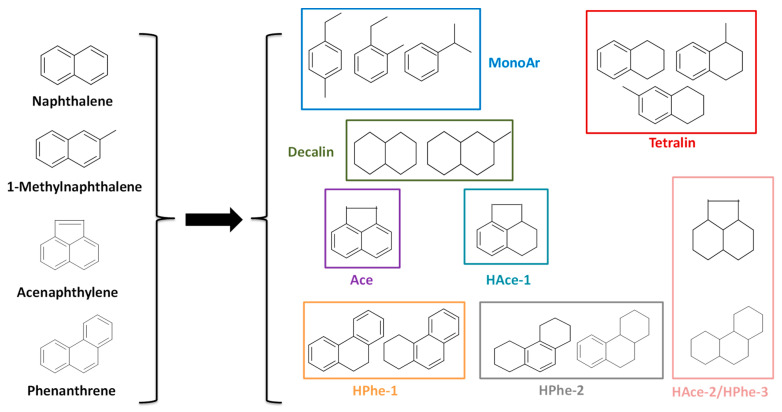
Chemical structure of the reactants and the different groups of obtained products.

**Figure 2 nanomaterials-11-02380-f002:**
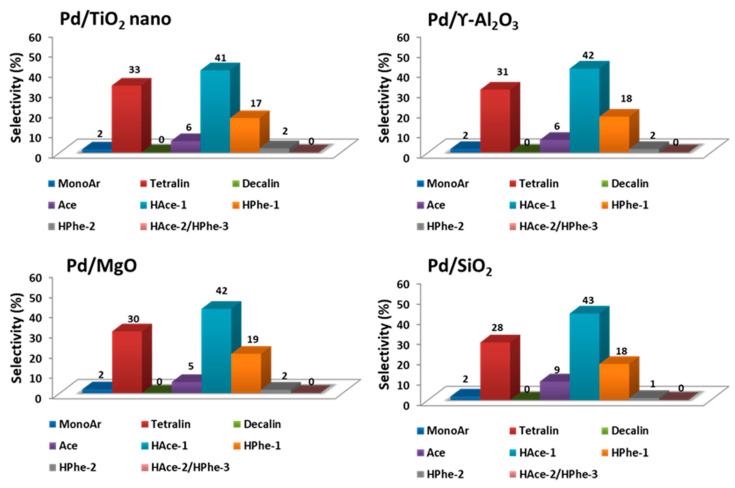
Selectivity to the different groups of products for metal oxides-supported Pd catalysts compared at 54–57% conversion. Reaction conditions: 0.5 g of tars-type compounds, 4 g of n-hexadecane, 0.2 g of catalyst at 250 °C and 30 bar of H_2_ during 7 h.

**Figure 3 nanomaterials-11-02380-f003:**
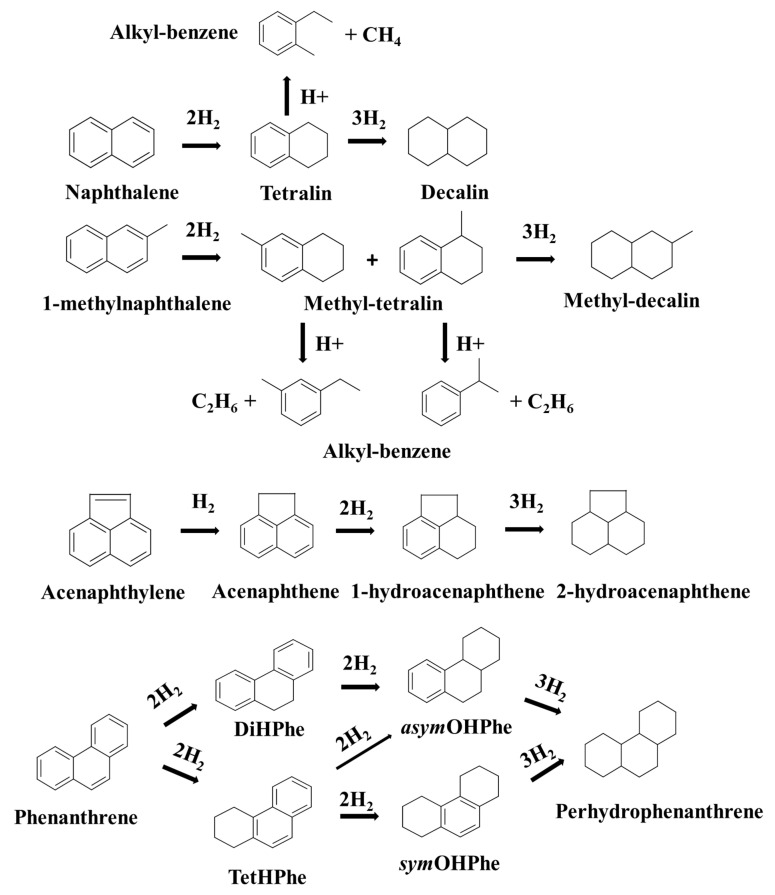
Proposed reaction scheme for the mild hydrotreatment of tars-type compounds.

**Figure 4 nanomaterials-11-02380-f004:**
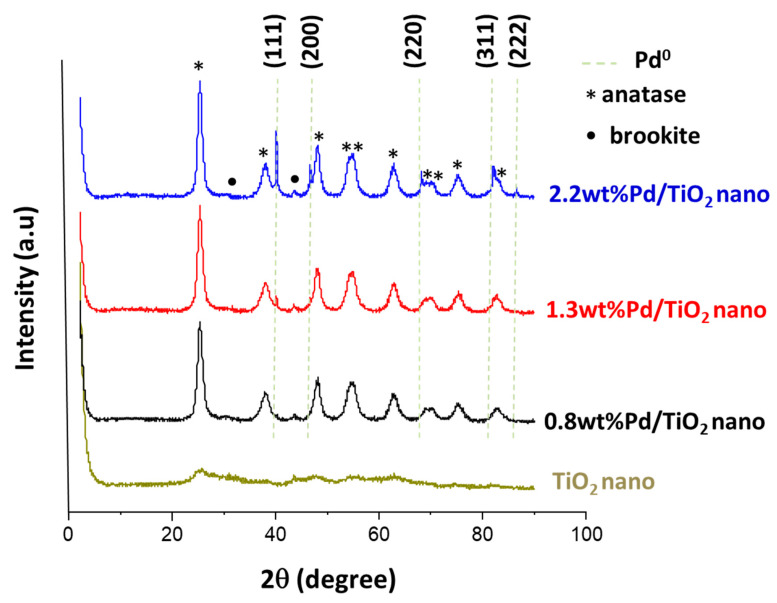
X-ray diffraction patterns for pure TiO_2_ nano and different TiO_2_ nano-supported Pd catalysts.

**Figure 5 nanomaterials-11-02380-f005:**
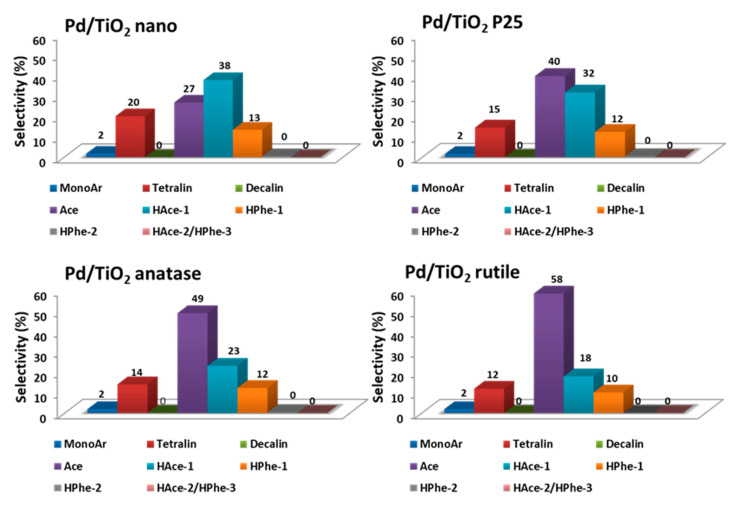
Selectivity to the different groups of products for TiO_2_-supported Pd catalysts at 35–40% conversion. Reaction conditions: 0.5 g of tars-type compounds, 4 g of n-hexadecane, 0.2 g of catalyst, at 250 °C and 30 bar of H_2_ during 7 h.

**Figure 6 nanomaterials-11-02380-f006:**
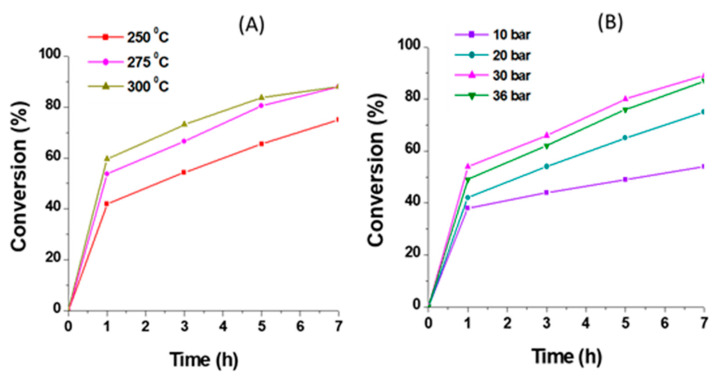
Conversion vs. time for 1.3 wt% Pd/TiO_2_ nano catalyst in tars mild hydrotreatment: (**A**) At 30 bar of H_2_, 0.2 g catalyst, during 7 h; (**B**) At 275 °C, 0.2 g catalyst, during 7 h.

**Figure 7 nanomaterials-11-02380-f007:**
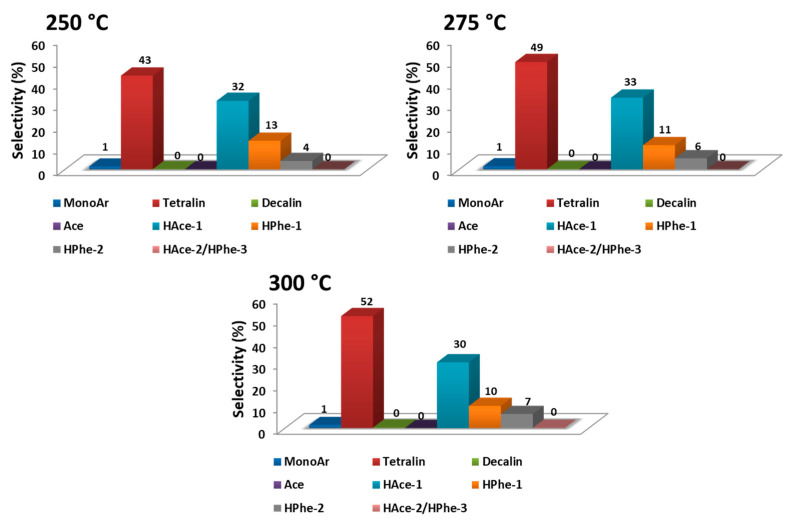
Selectivity to the different groups of products (compared at ≈75% conversion) for 1.3 wt% Pd/TiO_2_ nano by using different temperatures. Reaction conditions: 0.5 g of tars-type compounds, 4 g of n-hexadecane, 0.2 g of catalyst, at 30 bar of H_2_ during 7 h.

**Figure 8 nanomaterials-11-02380-f008:**
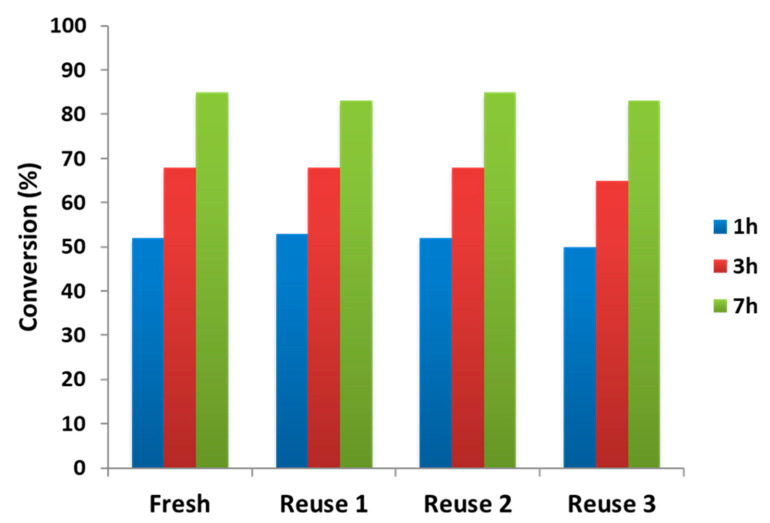
Reusability tests for Pd/TiO_2_ nano catalyst in the mild hydrotreatment of tars. Reaction conditions: 0.5 g of tars-type compounds, 4 g of n-hexadecane, 0.2 g of catalyst, at 275 °C and 30 bar of H_2_.

**Figure 9 nanomaterials-11-02380-f009:**
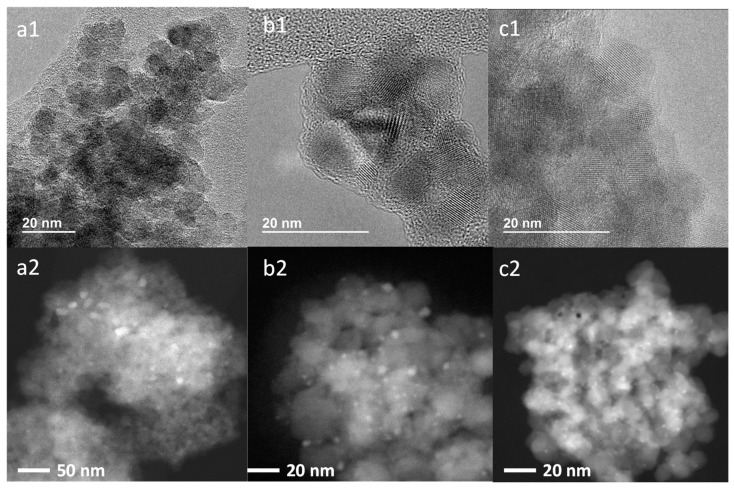
HR-TEM (1) and HR-STEM (2) images of (**a**) fresh, (**b**) reused and (**c**) regenerated (with H_2_) Pd/TiO_2_ nano catalyst.

**Figure 10 nanomaterials-11-02380-f010:**
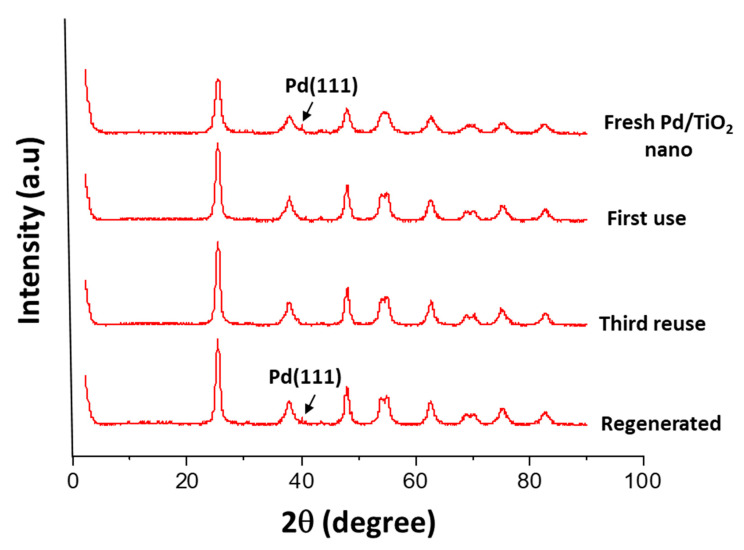
X-ray diffraction patterns of fresh, after uses and after regeneration (with H_2_) of Pd/TiO_2_ nano catalyst.

**Figure 11 nanomaterials-11-02380-f011:**
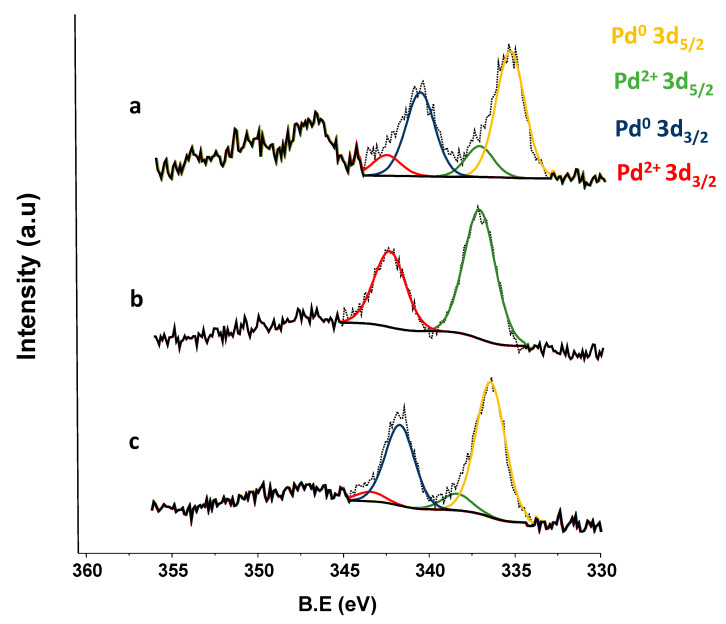
X-ray photoelectron spectroscopy (XPS) patterns of Pd 3d for (**a**) fresh, (**b**) after uses and (**c**) after regeneration (with H_2_) of Pd/TiO_2_ nano catalyst.

**Table 1 nanomaterials-11-02380-t001:** Physico-chemical properties of metal oxides-supported Pd catalysts and their catalytic performance in tars-type compounds mild hydrotreatment ^a^.

Catalyst	Pd (wt%) ^b^	Support Surface Area (m^2^/g) ^c^	Conversion (%)	Turn Over Number
Pd/TiO_2_ nano	2.3	162	89	67
Pd/γ-Al_2_O_3_	2.3	250	74	46
Pd/MgO	1.7	382	63	54
Pd/SiO_2_	2.0	505	54	35

^a^ Reaction conditions: 0.5 g of tars-type compounds, 4 g of n-hexadecane, at 250 °C and 30 bar of H_2_ during 7 h. ^b^ Pd contents measured by ICP. ^c^ Calculated from N_2_ adsorption isotherms (BET method).

**Table 2 nanomaterials-11-02380-t002:** Main textural, physico-chemical properties and catalytic performance in tars mild hydrotreatment for different TiO_2_-supported Pd catalysts ^a^.

Catalyst	Pd wt% ^b^	S_BET_ Support (m^2^/g) ^c^	Pd Dispersion (%) ^d^	Particle Size (nm) ^d^	Conversion (%)	TON (mols prod./mols Pd)
Pd/TiO_2_ nano	1.3	162	8.1	13	75	91
Pd/TiO_2_ P25	1.4	50	8.9	13	66	67
Pd/TiO_2_ anatase	1.4	11	2.3	49	44	46
Pd/TiO_2_ rutile	1.3	2	1.1	≥100	35	38

^a^Reaction conditions: 0.5 g of tars-type compounds, 4 g of n-hexadecane, 0.2 g of catalyst, at 250 °C and 30 bar of H_2_ during 7 h. ^b^ Measured by ICP; ^c^ Values calculated from N_2_ adsorption isotherms (BET method); ^d^ Average diameter of Pd nanoparticles and Pd dispersion calculated by CO chemisorption, with the stoichiometry being Pd:CO = 1:1.

**Table 3 nanomaterials-11-02380-t003:** Quantitative data for NH_3_-TPD and H_2_-TPR.

Catalyst	Total Amount of Acid Sites (µmol/g)	Total Amount of Adsorbed H_2_ (µmol/g)
Pd/TiO_2_ nano	367	193
Pd/TiO_2_ P25	200	115
Pd/TiO_2_ anatase	43	<1
Pd/TiO_2_ rutile	23	<1

**Table 4 nanomaterials-11-02380-t004:** Effect of reusability and regeneration with H_2_ on carbon deposition over Pd/TiO_2_ nano catalyst in the tars mild hydrotreatment.

Pd/TiO_2_ Nano	% N ^a^	% C ^a^	% H ^a^	Pd (wt%) ^b^	Pd Particle Size (nm) ^c^
Fresh catalyst	0.000	0.085	0.434	1.3	2.7
After first use	0.000	0.753	0.368	1.3	-
After third use	0.118	1.492	0.280	1.3	2.3
After regeneration	0.078	0.599	0.241	-	2.0

^a^ Results from elemental analysis (EA). ^b^ measured by ICP. ^c^ Average diameter of Pd nanoparticles calculated from TEM measurements of, at least, 100 particles.

**Table 5 nanomaterials-11-02380-t005:** X-ray photoelectron spectroscopy (XPS) values of binding energies for the fresh, reused and regenerated (with H_2_) Pd/TiO_2_ nano catalyst.

Pd/TiO_2_ Nano	Pd 3d_5/2_		Pd 3d_3/2_		% Pd^0^ 3d_5/2_	% Pd^2+^ 3d_5/2_
	Pd^0^	Pd^2+^	Pd^0^	Pd^2+^		
Fresh	334.64	336.48	339.87	341.84	80.98	19.02
Reused	-	335.71	-	340.96	-	100
Regenerated	335.07	337.05	340.35	342.05	89.23	10.77

**Table 6 nanomaterials-11-02380-t006:** Catalytic activity comparison between Pd/TiO_2_ nano and other commercial catalysts in the tars mild hydrotreatment ^a^.

Catalyst	Conv (%)	TON ^d^	Mass balance (%)	Product Selectivity (%)			% C
				MonoAr	Tetralin	HPhe-1/HPhe-2	
Pd/TiO_2_ nano	88	103	98	1.2	55.2	6.0/9.0	0.8
Pd/Al_2_O_3_ ^b^	69	97	91	1.4	37.4	11.9/3.1	0.9
Pt/Al_2_O_3_ ^b^	57	143	91	2.2	27.1	14.9/0.0	0.7
NiMoS/Al_2_O_3_ ^c^	93	7	94	1.3	54.3	5.2/9.1	2.0
CoMoS/SiAl ^c^	73	7	90	1.8	42.2	10.4/3.0	1.7
Pd/USY ^b^	98	85	54	5.5	33.6	1.3/3.4	10.0
Pt/USY ^b^	98	144	54	5.0	36.7	1.1/4.2	15.7

^a^Reaction conditions: 0.5 g of tars-type compounds, 4 g of n-hexadecane, at 275 °C and 30 bar of H_2_ during 7 h. ^b^ For Pd- and Pt-based catalysts, metal content ≈1.0 wt%, except for Pd/TiO_2_ nano (1.3 wt%). ^c^ In NiMoS/Al_2_O_3_ Ni content = 3.3 wt% and Mo content = 12.0 wt%; and in CoMoS/SiO_2_-Al_2_O_3_, Co content = 2.8 wt%; and Mo content = 8.0 wt%. ^d^ TON = mols of products/mols of Metal in catalyst.

## Data Availability

Not applicable.

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
