# Peer review of "TiO2-Supported Pd as an Efficient and Stable Catalyst for the Mild Hydrotreatment of Tar-Type Compounds"

_nanomaterials, 2021, doi:10.3390/nano11092380_

Round 1

Reviewer 1 Report

This manuscript reported that Pd supported on TiO2 with different crystalline phases for the mild hydrotreatment of tars-type compounds produced partially hydrogenated products. The authors pointed out that compared with commercial and previously reported catalysts, the Pd/TiO2 Nano material is efficient and stable catalyst for mild hydrotreating of tars-type compounds. However, there were still errors and something needed to be further improved. Thus, the manuscript cannot be considered for acceptance in this status.

  1. The “H2, N2” in the reaction conditions of Table 1 should be corrected. Other parts also had the same problems, which should be corrected.
  2. I strongly suggested that a bar chart like Figure 2 should be indicated exact values for readers to read.
  3. The standard PDF data for pure TiO2 should be added in Figure 4,Figure 11 and Figure S6.
  4. The N2 adsorption-desorption isotherms should be data mapping using Origin or Excel.
  5. The explanation of H2-TPR profiles should be express the test purpose more clearly.
  6. I suggested that the author should further proved that why TiO2 as a carrier is superior to other metal oxides through supplementary tests or references. Besides, the manuscript did not explain the action mechanism of catalyst in detail, so the further supplementary explanations should be given.
  7. The catalyst still showed high performance in the third cycle. At this time, the cycle experiment should be continued until the catalytic performance decreased significantly.

Reviewer 2 Report

Nanomaterials-1337028

Title: TiO2-supported Pd as an efficient and stable catalyst for the mild hydrotreatment of tars-type compounds

Authors: Zaher Raad, Joumana Toufaily, Tayssir Hamieh, Marcelo E. Domine

The authors study the mild hudrogenation of model mixture of tars-type compounds with Pd/TiO2 catalysts. The acidity and the surface are of the catalysts favors catalytic activity. The optimum loading was found 1.3 and the maximum conversion were encountered at 275 °C, 30 bar of H2, 0.2 g of catalyst during 7 h reaction. Pd/TiO2 nano catalyst was found to be stable, more active than other commonly used metal/alumina, and more selective than metal/USY zeolites

The work is well performed and it can be accepted for publication after revision.

I don’t agree that 1.3Pd/TiO2 is the most active catalyst. I understand the authors that this catalyst has a mean activity and TON but it is not the most active

Why the Pd/TiO2 anatase has such a low SSA? I was expected much higher SSA. Please comment

The catalyst supported on P25 has a high dispersion (equal with TIO2 nano) although the SSA is about the 1/3 of the other titania. According to your data the Pd crystallites can be form at abut 2% Pd loading in TiO2 nano. But no crystallites can be observed in P25 ? why? Please comment.

Please add in abstract the physicochemical characterization

Why the Nano… is with capital N? Please correct

I am not sure that conversion and selectivity have units. Please erase mol

Figure 5 can be deleted.

Although I agree with the authors about selectivity, strictly speaking the selectivity should be measured under the same conversion, (for example partial pressure of H2 may be different concentration of tar compounds may be different etc) and then the results can be different. Please rephrase this part.

Why after reaction (with 30 bar H2) there is oxidation of Pd(0) to (Pd(II)? Please explain and as the XPS clearly shows this oxidation is fully.

Minor comments

Please check for typos and refine English in some parts.

Figure 11 axis title 2θ not 2O

Round 2

Reviewer 1 Report

The author modified the manuscript point by point in accordance with the comments. I recommend the manuscript could be accept after a minor revision. Fig. S15 and Fig. S16 should provide the complete N2 adsorption and desorption data for a N2 adsorption-desorption curves.

Reviewer 2 Report

The authors revised their manuscript according to my suggestions

Now their manuscript can be accepted for publication

Author Response

Many thanks for all your comments and suggestions that certainly help us to improve the quility of our research work.